# Enhancing Prostate Cancer Diagnosis with a Novel Artificial Intelligence-Based Web Application: Synergizing Deep Learning Models, Multimodal Data, and Insights from Usability Study with Pathologists

**DOI:** 10.3390/cancers15235659

**Published:** 2023-11-30

**Authors:** Akarsh Singh, Shruti Randive, Anne Breggia, Bilal Ahmad, Robert Christman, Saeed Amal

**Affiliations:** 1College of Engineering, Northeastern University, Boston, MA 02115, USA; singh.akar@northeastern.edu (A.S.); randive.s@northeastern.edu (S.R.); 2Maine Health Institute for Research, Scarborough, ME 04074, USA; 3Maine Medical Center, Portland, ME 04102, USA; bilal.ahmad@spectrumhcp.com (B.A.); robert.christman@spectrumhcp.com (R.C.); 4The Roux Institute, Department of Bioengineering, College of Engineering, Northeastern University, Boston, MA 02115, USA

**Keywords:** prostate cancer, digital pathology, artificial intelligence, multimodal data, human computer interaction, usability test, biopsy grading, clinical validation

## Abstract

**Simple Summary:**

Prostate cancer is a significant cause of male cancer-related deaths in the US. Checking how severe the cancer is helps in planning treatment. Modern AI methods are good at grading cancer, but they are not used much in hospitals yet. We developed a new online tool that combines human expertise and smart AI predictions to grade prostate cancer. Experienced doctors helped us improve the tool by answering questions in a survey and a test called the NASA TLX Usability Test. This helped ensure that the tool meets the needs of intended medical users and proves a valuable addition in clinics. The feedback obtained was positive for three themes that included acceptability, ease of use, and understanding with suggested application feature improvements as feedback for real world hospital use. The task completion process was not very demanding overall. The web application has potential to serve well upon minor improvements based on feedback.

**Abstract:**

Prostate cancer remains a significant cause of male cancer mortality in the United States, with an estimated 288,300 new cases in 2023. Accurate grading of prostate cancer is crucial for ascertaining disease severity and shaping treatment strategies. Modern deep learning techniques show promise in grading biopsies, but there is a gap in integrating these advances into clinical practice. Our web platform tackles this challenge by integrating human expertise with AI-driven grading, incorporating diverse data sources. We gathered feedback from four pathologists and one medical practitioner to assess usability and real-world alignment through a survey and the NASA TLX Usability Test. Notably, 60% of users found it easy to navigate, rating it 5.5 out of 7 for ease of understanding. Users appreciated self-explanatory information in popup tabs. For ease of use, all users favored the detailed summary tab, rating it 6.5 out of 7. While 80% felt patient demographics beyond age were unnecessary, high-resolution biopsy images were deemed vital. Acceptability was high, with all users willing to adopt the app, and some believed it could reduce workload. The NASA TLX Usability Test indicated a low–moderate perceived workload, suggesting room for improved explanations and data visualization.

## 1. Introduction

Prostate cancer (PCa) is the most prevalent form of cancer among males in the United States. According to data from the National Institute of Health (NIH), there were 268,490 new cases of PCa, and 34,500 PCa-related deaths reported in 2022. Recent advancements in deep learning have shown promising results for diagnosing prostate cancer in biopsies [1]. In their work, Refs. [2,3,4,5,6,7] demonstrated that computational pathology and modern deep learning techniques can achieve expert-level pathology diagnosis for prostate cancer detection. However, it is crucial to ensure effective communication and interpretation of these ideas by pathologists to aid in cancer-related decision-making processes.

To address this challenge, authors of [8] developed Paige Alpha, a convolutional neural network (CNN) based on the multiple instance learning algorithm. Authors simulated a reduction of 65.50% in diagnostic time for cancer grading and reporting. An earlier version, [9] the Paige Prostate Alpha application, improved the prostate cancer detection sensitivity of pathologists by 16%. This integration of human expertise and AI demonstrated the best performance, highlighting the invaluable contributions of human experts in AI-driven healthcare applications. Both these studies used whole-slide images (WSIs) of core needle biopsies, similar to imaging data we used. However, these studies have limitations. AI capabilities are limited to cancer detection (not grading), limited diversity in datasets used, and only 2–3 pathologists for validation. The studies also do not fully replicate the real-world diagnostic workflow, where pathologists often use multiple sources of information. Alternatively, AI-aided applications for automatic segmentation on prostate MR images were also identified. Many studies [10,11,12,13] have suggested that AI-based computer-aided detection systems have potential clinical utility in clinically significant prostate cancer (csPCa) detection. Authors of [14] segmented prostate outer contour and peripheral zone contour using cascaded CNNs. Dice similarity coefficient for manual outlining vs AI-generated segmentations was 92.7 ± 4.2% for the total whole prostate gland, using diffusion-weighted images from 163 subjects. Authors of [15] used this proprietary software (https://insightsimaging.springeropen.com/articles/10.1186/s13244-023-01421-w/figures/1; accessed on 3 November 2023) in their study for detecting csPCa (480 multiparametric MRI) with 16 radiologists from four hospitals. They concluded that AI-aided detection reduced the median reading time of each case by 56.3%

In our recent work [16], we introduced a novel interactive application designed to resonate with clinical workflows. The application leveraged digitized prostate tissue biopsies to predict ISUP grades with high accuracy, employing EfficientNet-B1 [17]-based deep learning models. The application included interactive visualizations for transparent model predictions, highlighting potential cancerous regions. This aided pathologists in grading and reduced their workload. Positive feedback from three expert pathologists emphasizes the relevance of our AI web application for optimizing digital pathology workflows.

Our current study reviews crucial aspects of successful healthcare AI applications, focusing on the enhancement of our scalable web application for prostate cancer detection and diagnosis. We posit that the integration of diverse data modalities, encompassing patient EHR data, high-resolution biopsy WSIs, and report summaries alongside advanced grading predictions, will lead to optimization of clinical workflow for prostate cancer grading. Recent advances in AI have demonstrated the active use and popularity of multimodal data sources [18,19]. Additionally, multimodal data provide better interpretability by analyzing the contributions of each modality to the final prediction [20]. Our objective is to create an application that seamlessly integrates into the prostate cancer diagnosis workflow. It is tailored to the needs of medical professionals through ongoing communication and feedback in the form of surveys and usability tests. Our approach draws inspiration from a study [21] that developed machine learning models for diagnosing peripheral artery disease. The study highlighted the effectiveness of automated tools, dashboards, and visualizations in enhancing usability.

To explore the relevance of applications in conveying valuable information and insightful findings, it is equally important to assess user feedback and experience for a well-adapted and personalized application. Surveys emerge as an effective tool for this purpose, allowing for the improvement of user experience and usability of web applications. They help identify pain points, gather feedback on specific features and functionality, and enable informed decision-making for enhancing the application to better meet user needs [22,23]. For instance, authors of [24] used a survey to understand the role of experiential features in the success of well-known products, proposing an interpretative model for marketing managers to enhance customer experience.

Furthermore, surveys have been instrumental in identifying concerns related to the use of AI in healthcare. Authors of [25] conducted a survey to categorize technological, ethical, and regulatory concerns, which significantly influence individuals’ assessment of the risks and benefits associated with AI-based clinical decision support systems. Another survey [26] aimed to understand participants’ perceptions of AI in healthcare, revealing positive views due to its potential to improve efficiency and reduce healthcare costs. However, concerns were raised regarding data privacy, patient safety, and technological maturity. Suggestions for improvement included enhanced personalization, customizability, user experience, design, and interoperability with other electronic devices.

Usability tests are of paramount importance in cancer applications. For instance, authors of [27] aimed to develop and assess a smartphone application for prostate cancer screening based on the Rotterdam Prostate Cancer Risk Calculator (RPCRC) app. Their usability assessment, involving 92 participants, including urologists, general practitioners, and medical students, yielded high scores in system usefulness (92%), information quality (87%), and interface quality (89%). The study leveraged the IBM Post-Study System Usability Questionnaire (PSSUQ) to gather valuable quantitative data, showcasing robust usability measures among the 92 users. In another instance, authors of [28] created and evaluated a user-friendly medical chatbot, the prostate cancer communication assistant (PROSCA), to provide patient information about early prostate cancer detection. This chatbot, developed in collaboration with Heidelberg University Hospital and the German Cancer Research Center, was tested on 10 suspected prostate cancer patients alongside standard physician education. Usability and feedback were collected through questionnaires, revealing that 78% of patients found the chatbot easy to use without assistance, with 89% gaining substantial information about prostate cancer. It is essential to note the study’s limitations, including a small sample size of German-speaking men in Germany with prescheduled urology consultations. Furthermore, authors of [29] developed an artificial neural network tool for predicting locoregional recurrences in early-stage oral tongue cancer. The study involved 16 participants from various backgrounds and employed the System Usability Scale (SUS) to measure usability. Results from the SUS indicated that most participants did not require extensive learning or technical support (81.8%) but lacked confidence (36.4%). The SCS results demonstrated agreement on the model’s soundness (54.5%). The study underscored the need for post hoc explanations for web-based prognostic tools when SCS scores fall below 0.5. It is important to acknowledge the limitations of the study, including the lack of standardized methods for measuring the quality of explanations and a relatively small participant pool, which may limit generalizability. Metrics may not comprehensively encompass the multifaceted aspects of usability.

In our study, we address these limitations by carefully selecting unbiased participants for usability testing and opting for standardized usability tests to ensure the feedback, both quantitative and qualitative, holds merit. Furthermore, we strive to clarify the definitions of key terms in the context of ML models, ensuring that participant understanding of model results is thoroughly evaluated. Our research contributes to enhancing AI usability in clinical prostate cancer detection by improving visualization and integrating relevant EHR data. We have evaluated usability through surveys, utilizing diverse question formats and the NASA TLX Usability questionnaire. Our study focuses on quantifying the advantages of clinical tool adoption, particularly in terms of usability, information value, and user workload, as assessed by experts.

## 2. Materials and Methods

### 2.1. Dataset and Model Implementation Details

Exploration of our research paradigm stared with data accusation for model training. To provide a robust foundation for our research initiative, we commenced the training of our deep learning models by utilizing the Prostate Cancer Grade Assessment Challenge (PANDA) dataset. This dataset was sourced from the open-source deep learning competition hosted on Kaggle [30] and chosen because this competition represents a significant milestone, offering a comprehensive collection of 12,625 whole-slide biopsy images, making it the most extensive dataset of its kind. In support of model training for predictive analytics using the dataset and subsequent web application development, computational resources provided by Discovery (Massachusetts Green High Performance Computing Center, Holyoke, MA, USA), a high-performance computing facility catering to Northeastern University’s research community, were leveraged. Discovery facilitated the utilization of software applications capable of seamless execution in web browsers, obviating the need for additional configuration. Allocation of a compute node with specified cores and memory facilitated these operations, with our project directory allocated 750 GB of storage space. Utilizing the Kaggle API with secure credentials, we downloaded the extensive 411 GB PANDA dataset, comprising biopsy whole-slide images (WSIs) along with associated image masks and grade label tabular information for training (12,625) and testing (940). Model training and web application development were conducted using a VS code server equipped with an NVIDIA Tesla A100 GPU (CUDA cores: 6912; tensor cores: 432 for AI workloads; memory: 40 GB).

To enhance training performance, preprocessing involved the removal of redundant white space from biopsies, accomplished using the tiling algorithm published on Kaggle. This approach entails segmenting WSIs into patches, selecting the first N patches based on the maximum number of tissue pixel values, and combining augmented tiles into a single image. The resulting image is resized to fit the model input, adopting a dimension of 1024 × 1024.

For model architecture, we employed a 5-model ensemble methodology based on the EfficientNet-B1, a state-of-the-art convolutional neural network (CNN) architecture [17]. The EfficientNet framework incorporates compound coefficient scaling, optimizing model size with a penalty mechanism for computational and inference time, thereby ensuring balanced scaling across width, depth, and image resolution. Pretraining on ImageNet data [31] provided feature representations, followed by fine-tuning and training on three folds, validation on one fold, and testing on the last fold (total 5 folds). The average prediction of all 5 models was employed in the web application.

The adapted architecture for the multiclass classification problem of 6 classes included the replacement of the fully connected layer, subsequent to which a flatten layer and squeeze-and-excitation (SE) block were introduced for capturing channel-wise dependencies and recalibrating feature responses. This was followed by a dropout layer and a dense layer with SoftMax activation, respectively. BCEWithLogitsLoss served as the chosen loss function due to its robustness in one vs. all classification scenarios and robustness to over/underflow. Hyperparameter values are provided in Table 1 below. It is to be noted that other design considerations and architecture details are similar to the original EfficientNet architecture.

Importantly, the final model demonstrated significant efficacy, achieving an impressive quadratic weighted kappa agreement of 0.862 (95% confidence interval (CI), 0.840–0.884) on the external validation dataset sourced from the United States. Notably, our model attained state-of-the-art accuracy on the aforementioned dataset.

### 2.2. Web Application Development and Integration

To facilitate user interaction and results provision, a comprehensive web application was developed utilizing the Flask (Python) framework, thereby enabling facile development and seamless integration across various systems, leveraging the same computational resources as outlined in Section 2.1. The objective was to empower users to upload biopsy images and access thorough visualizations intertwined with electronic health record (EHR) data. In the absence of authentic EHR data, a method employing synthetic records was utilized to simulate patient profiles relevant to prostate cancer research, drawing insights from online reports from the National Institutes of Health (NIH) and consultations with pathologists. This synthetic dataset encapsulated critical elements, including patient demographics, medical history, and treatment details, while maintaining clinical realism in alignment with prevailing medical themes. Despite the relevance of the EHR data sample, its direct linkage to corresponding biopsy images for the same patient was unavailable to us. The primary objective of our study was to evaluate the practical utility of such data in conjunction with biopsy and artificial intelligence (AI)-generated grade predictions. Additionally, our application sought to elucidate the potential of integrating diverse data modalities to provide pathologists with a comprehensive diagnostic framework.

The image name, excluding the .tiff extension, was selected as the primary key, serving as a universally unique identifier for each row of generated EHR data in our database. This choice was informed by the uniqueness of image names in the PANDA dataset. The image upload process initiated as users transmitted POST requests, with the Flask route dedicated to handling uploads securely storing the image in a designated repository on the server. The image then underwent processing through our deep learning pipeline, a Python script designed to process uploaded images and employ saved ensemble models to yield predictions. The output included ensembled grade predictions and visualizations. The unique identifier associated with each image ensured a seamless correlation between images and their corresponding EHR data. Retrieval of EHR data, a pivotal component of the application, involved a secure database connection, facilitating access to patient information.

The establishment of interactive tabs was achieved through a synergistic combination of PostgreSQL queries and the Bootstrap framework. Leveraging the capabilities of the Flask web framework, Python code was employed to execute optimized PostgreSQL queries using the psycopg2 library. These queries were instrumental in retrieving patient-specific EHR information, thereby furnishing the requisite data for populating the interactive tabs. HTML was utilized to structure the content of the tabs, while CSS played a role in styling and formatting visual elements, ensuring a unified design. The Bootstrap framework played a pivotal role in enhancing the functionality of the interactive tabs, seamlessly integrating them into the user interface. The grid system provided by Bootstrap facilitated the layout of the tabs, ensuring consistency across diverse devices. Biopsy visualizations were managed using OpenCV. By amalgamating EHR-based visualizations with uploaded biopsy images, the application provided a comprehensive interface, enabling medical professionals to establish correlations between patient information and biopsy data.

In this configuration, scalability was meticulously considered, ensuring the application’s adaptability to large datasets and user numbers. Notably, the application exclusively functions within the Northeastern VPN, mitigating unauthorized use and ensuring security. To maintain a clean and scalable codebase, components and files were modularized. SQL queries, though not intricate, could be optimized using standard practices. Containerization was implemented to facilitate dynamic scaling based on demand and changing requirements. To mitigate the load time of static files such as CSS/JavaScript, the flask-compress extension was employed, and the response time for deep learning operations improved by 40% when GPU acceleration was available. Given the specific requirements and the understanding of limited requests in this initial application version, external plugins for features like performance monitoring, load testing, distribution, and balancing were deliberately limited. Future iterations of the application will leverage cloud services, as expounded upon in the conclusion section.

Challenges encountered during the development of the application were constrained. Flask, being less intricate compared to frameworks like Django, facilitated a smoother development process. The application comprised only two pages—a landing page and a result page following a successful biopsy upload—utilizing a simplified template, thereby contributing to ease of management. Minor bottlenecks during development manifested in the form of error handling, installation of specific dependencies, and the management of concurrent requests. However, it is pertinent to note that the application, in its current form, adheres to a relatively simplified paradigm compared to contemporary standards, making it accessible to individuals with a few years of software development experience.

### 2.3. Web Application Interface

Figure 1 and Figure 2 illustrate the web application’s user interface, portraying the interface in its pre- and post-biopsy upload states, respectively. In Figure 2, the left donut chart displays the final ISUP grade prediction derived from ensembled results. Given that ISUP grades [32] are whole numbers within the range of 1 to 5, the donut chart represents the likelihood of the 2 nearest grades. This specific design approach streamlines the focus for clinical pathologists, allowing them to initially examine patterns linked to the most relevant grades according to the deep learning models and potentially reduce the time spent on grading. On the right side, the donut chart provides an instance of the SoftMax output from a random model (1 out of 5), yielding a probability distribution that signifies the probability of the biopsy corresponding to each grade, based on the model’s outcomes. The chart’s title includes the argmax, denoting the grade with the highest probability assigned by the model for the image. The color spectrum chosen comprised 6 distinct colors ranging green, showcasing benign/no cancer; yellow-orange for showcasing intermediate cancer grades like ISUP 1, 2, 3; and reds for severe grades like ISUP 4 and 5. The color gradient was chosen for as it is self-explanatory and intuitive (Appendix A show web-layout of the application before and after uploading biopsies with multiple hovers). Donut charts with further tabs are an example of multifaceted visualizations. Multifaceted visualizations provide a comprehensive view of complex data, aiding pattern recognition and informed decision-making [33]. They reduce cognitive load, simplify data exploration, and enhance communication by presenting multiple dimensions in a single view, making them valuable across various domains [34]. Their customizability and efficiency in conveying rich data make them a preferred choice for data analysis and interpretation.

Additionally, the web application incorporates four interactive tabs, each providing pertinent information that underpins the decision-making process, as depicted in Figure 3. The “Biopsy” tab furnishes a visual representation of the whole-slide images (WSIs), enabling users to directly assess the image for cancer grading using their own devices. Should electronic health record (EHR) data be available for the given biopsy, the “Demographics” and “NLP Summary” tabs are generated. It is to be noted that NLP on prescriptions was not performed in real time or using a deep learning algorithm. The purpose was to test if such information was valuable for our purpose. Within the “Demographics” tab, demographic particulars regarding the patient, such as height, weight, and body mass index (BMI), are presented. The “NLP Summary” tab consolidates essential clinical observations from recent visits, endowing decision makers with supplementary insights into the patient’s medical background, allergies, treatments, and existing conditions.

Lastly, as portrayed in Figure 4, in the “Detailed Summary” tab (application view), an inclusive combination of previously cited data is presented. This encompasses specific risk factors (e.g., preexisting hypertension), presented through a static dashboard format. This suite of interactive tabs is meticulously designed to highlight and convey the most pertinent information, augment the depth and quality of data, and refine visual representations wherever feasible. In doing so, an elevated interactive experience is curated for users, fostering more comprehensive and meaningful engagement. Appendix A illustrate the appearance of each tab upon direct access from the application.

### 2.4. Experimental Objectives

This study aimed to assess the feasibility of our application within clinical practice. It also aimed to evaluate user feedback from pathologists and other medical professionals who have integrated the application into their workflows. Experts provided valuable feedback. Additionally, a comprehensive review of relevant research on integrating multi-modal data into healthcare-related AI models unequivocally substantiated the pronounced advantages of harnessing diverse knowledge sources to improve decision-making processes. The application was developed to harness a spectrum of medical data modalities, including annotated biopsy regions catering to meticulous inspection, computer-enhanced whole-slide images (WSIs) primed for digital assessment, risk score visualization complemented by performance metrics from AI models, natural language processing (NLP) summaries, and patient demographic particulars. These diverse elements were amalgamated within our interactive web-based framework, as expounded upon earlier, specifically attuned to the nuances of prostate cancer detection and diagnosis. To build on the previous work [16], this study primarily focuses on the following improvements:

Firstly, optimizing the assimilation of assorted healthcare data modalities, spanning electronic health records (EHRs), risk indicators, demographic variables, and insights distilled from NLP-derived prescription and report summaries. Noteworthy is the augmentation of our application’s visual aesthetic, paralleled by an expanded reservoir of patient data, aiming to quantify the usefulness to pathologists during the cancer grading process.

Secondly, a comprehensive survey was meticulously designed to evaluate the effectiveness, clarity, and visual appeal of our application’s presentation and usability. The survey consisted of six primary tasks with multiple sub-questions, inspired by the NASA (National Aeronautics and Space Administration) TLX Usability Test [35], completed by partner pathologists who thoroughly used the application and provided suggestions for enhancements. The questionnaire, aligned with application features, emphasized clinical adaptation, user experience (especially for nontechnical medical professionals), and personal preferences in prostate cancer diagnosis workflows.

### 2.5. Survey Outline

The survey was meticulously crafted to gather comprehensive feedback from medical practitioners and pathologists who used the web application. Our survey design is best characterized as a descriptive survey, a well-established approach in social sciences and research. This type of survey focuses on collecting data to describe a population or phenomenon, providing a snapshot of the present conditions and gathering insights into the characteristics, preferences, and opinions of participants. In this context, our survey aims to offer a comprehensive description of the usability, accessibility, and acceptability of our prostate cancer web application among pathologists and medical professionals. Through a structured questionnaire, we gathered data to outline the users’ experiences and preferences regarding the application’s design and functionality. Ref. [36] proved to be a handy resource to guide our survey design process. It directed us to choose the appropriate survey design options and minimize unconscious bias.

The survey underwent a rigorous co-design process that included identification of themes for assessment, prototype questionnaire design, consultation with experts, and iterative improvements, followed by pilot testing. Our questionnaire was divided into six primary sections, aiming to collect detailed input on each aspect of the web application, as summarized in Table 2 below.

The questions were thoughtfully designed to evaluate usability, clarity, accessibility, and acceptability. By employing a descriptive survey, our research aligned with established methodologies in the field. All rating-based questions were on a Likert scale [37] ranging from 1 to 7. It allows us to comprehensively document user experiences, understand their needs, and derive insights that will inform the enhancement of our web application.

To comprehensively evaluate the post-application workload, our survey incorporated six questions that encompassed the NASA Task Load Index (TLX) usability test. The NASA TLX is a well-established, subjective workload assessment instrument that has attained widespread recognition as a preferred choice in a spectrum of industries and scholarly investigations. This test enhances usability assessment by considering various dimensions such as mental, physical, and temporal demands, performance, effort, and user frustration experienced when interacting with a system (on a scale of 1 to 20, where lower score is considered better), in this case, our web application [38]. Historically, this test has proven beneficial for researchers in multiple domains since its inception in the late 1970s, ensuring consistent data collection and a user-centric approach based on individual perspectives and subjective experiences [39]. Various studies have adopted this approach to assess process improvements and workloads for healthcare procedures and clinical practices [40,41,42]. Additionally, the TLX test enables comparison between different system versions, making it valuable for iterative design and examination, incorporating both qualitative feedback and quantitative data.

To assure its effectiveness, the survey underwent a rigorous validation process after completing the first draft, involving consultation with experts in human–computer interaction (HCI). It was also pilot tested on a small sample 4 healthcare research students to detect any potential issues or ambiguities. Iterative adjustments were introduced to specific questions, and contextual elements pertinent to the realm of artificial intelligence (AI) were judiciously incorporated based on feedback. This refinement became particularly necessary as it was discerned that certain aspects of the output visualizations required enhanced comprehension. Furthermore, the survey’s format and structure were optimized to enhance clarity and user-friendliness. The survey, which also included TLX questions, was administered through an online platform, was designed to be time-efficient, with the duration for completion falling within a range of 7 to 12 min. The delivery method involved an invitation-access online form to ensure ease of participation and data collection.

### 2.6. Cohort

A panel of 4 experienced pathologists and 1 medical expert, carefully selected for their domain knowledge, engaged with the application based on concise instructions as users. Subsequently, they provided feedback by completing a survey. It is noteworthy that these expert users hailed from distinct medical centers, specifically Maine Health and Spectrum Healthcare with a combined 50+ years of medical experience dealing with prostate cancer diagnosis. This deliberate selection of participants from varied backgrounds enhances the robustness of our evaluation process, which took slightly over 3 weeks to distribute and complete.

## 3. Results

The survey was designed in a way to analyze three themes related to our web application: ease of understanding, ease of use, and acceptability. Our baseline objective was to evaluate our applications capacity to garner a consensus among the majority of participants regarding all the underlying themes. Furthermore, we aimed to ascertain the application’s validity and its potential to serve as a justifiable tool for real-world clinical practice. To complement these aspects, we also incorporated findings from the TLX Usability Test, allowing us to quantitatively evaluate workload metrics associated with user interactions with the web application and the process of survey completion. This comprehensive approach not only provides a nuanced understanding of user perspectives with descriptive feedback but also offers a quantitative assessment of their experiences.

### 3.1. Ease of Understanding

We could quantify that 3 out of 5 (60%) users were comfortable navigating through the web application and interpreting AI predictive results along with processing additional patient information provided. Since all participants could be considered a nontechnical audience for AI result interpretation, a brief explanation of what the donut charts signified aided most users in clearly understanding what the visualizations intended to portray. The aspect of ease of understanding, both in terms of information presented and intuitiveness, garnered an impressive average rating of 5.5 out of 7 on a Likert scale. It is noteworthy that 40% of users required a brief minute to grasp the concept, despite a functional explanation. The average consensus regarding initial frustration in understanding the results was moderate at 4 out of 7, with most users concurring. Furthermore, it is essential to highlight that all users unanimously expressed strong agreement regarding the self-explanatory nature of the additional information presented through the four popup tabs (biopsy tab, demographics tab, NLP summary tab, detailed summary tab), which received an average rating of 6 out of 7. All users expressed that the information provided could be easily read and comprehended.

However, it is noteworthy that 2 out of 5 users encountered challenges in understanding the right donut chart, which denotes the probabilities associated with each ISUP grade for the uploaded H&E stained WSI (refer to Figure 2). One user provided descriptive feedback, highlighting that this representation lacked intuitiveness and user-friendliness, suggesting that the option to manually enable this representation should be considered. Both of these users expressed their confusion, primarily stemming from fractional predictive output observed in the left donut chart (visible in Figure 2). In contrast, the remaining three users could effectively interpret the probability distribution displayed in both donut charts, perceiving it as a valuable transparency feature revealing how our model generated predictions, thereby enhancing trust in the system. Minor suggestions by two users were provided to improve the visual appeal of the web-application, whereas three had no inputs to increase the value potency of our application deeming to be fit for the intended purpose of cancer diagnosis.

### 3.2. Ease of Use

There was a 100% consensus among all users regarding strong preference for the detailed summary tab, which thoughtfully showcased all information into one comprehensive view. This preference was unmistakable, as evidenced by an average rating of 6.5 out of 7. The invaluable feedback collected indicated that this presentation style not only streamlined their workflow but also significantly expedited the process, an improvement noted by 60% of users. Additionally, 80% expressed that this approach reduced their overall effort when evaluating the entirety of the information simultaneously. A noteworthy advantage of allowing users to interact with each tab initially was the mitigation of any primacy bias, ensuring a more objective consensus. It is worth noting that 80% of users indicated that, apart from age, patient demographic information did not significantly contribute to prostate cancer diagnosis. Some suggestions included incorporating additional datapoints like family history of prostate cancer, case number, and patient name to enhance the diagnostic process. Moreover, 80% of users emphasized the importance of higher-resolution biopsy images for more meaningful insights, reinforcing the significance of image quality in the diagnostic workflow.

Three out of five users expressed that summarization of reports using NLP techniques was not useful for diagnosis. One user said that the information could be distracting. Two users provided us with feedback to clearly mention the specifics of the type of digital file to be uploaded as a biopsy. Nevertheless, all users agreed that navigating through different sections of the application was straightforward.

### 3.3. Acceptability

In a comprehensive assessment, the web application emerged as highly promising for seamless integration into users’ clinical workflows, a unanimous sentiment shown by all participants (100% agreement). Notably, five out of five users expressed a willingness to adopt the application, contingent on access to the highest biopsy resolution. Moreover, a substantial majority, or three out of five users, recognized the potential for the application to reduce mental exertion and time consumption in their clinical routines. Three out of five users firmly asserted that the predictive results would introduce no bias into their decision-making process. Conversely, two out of five users acknowledged that, given the AI model’s demonstrated accuracy, they would actively utilize the results to navigate their decision-making process, emphasizing a preference for inspecting biopsies for patterns closer to the predicted grade initially. Valuable user feedback included recommendations for alternative chart styles, with two out of five users suggesting that a vertical bar chart might prove more effective for their needs. Additionally, concerns were raised about the potential ambiguity in the color spectrum employed (green–yellow–red) to represent grade severity, potentially leading to misinterpretations by one participant. However, all users unequivocally commended the effective use of chart legends (6/7 rating), representation clarity (5/7 rating), and the hover percentage feature, which served as a useful tool for cross-checking results (6/7 rating).

In the final segment of the evaluation, seven concluding questions gauging the web application’s acceptability and overall accessibility garnered an average positive rating of 5.5 out of 7, underscoring its potential as a valuable tool in the healthcare domain.

### 3.4. TLX NASA Usability Test

In the final round of evaluation, six questions were included to assess the perceived workload or mental effort associated with using the application and completing the survey. This test was selected for its robustness and wide-scale acceptability in academia and industrial practices. Participants rated these tasks on a scale of 1 to 20, with lower scores indicating a more favorable experience or lower perceived demand. The results revealed that the categories inducing the most demand, on average, were frustration level and mental demand, both receiving a mean score of 35%, indicating a below intermediate level of perceived demand. The highest median scores were observed for mental demand (45%) and temporal demand (50%), categorizing them as having a medium level of difficulty. Conversely, the categories associated with minimal workload were physical demand for task completion (mean: 17.5%; median: 20%) and effort level (mean: 22.5%; median: 20%). None of the categories leaned towards high difficulty or higher demand, with all ratings falling below 70%. This suggests that users did not perceive any task as particularly challenging or demanding. We exhaustively searched for baselines to validate our TLX results. There is no gold standard for “good performance”, which is context-dependent. What is considered good performance for one task or system may not be the same for another. The same has been corroborated by other scientific studies [43].

More details regarding average and median scores for all categories are provided in Table 3 below.

Table 4 presents more specific descriptive feedback provided on the abovementioned themes (in Section 3.1, Section 3.2 and Section 3.3) as well as additional themes. Based on qualitative feedback received, we could further divide feedback into subthemes and provide an overall assessment based on their feedback and experiences. Specific feedback would help us further fine-tune our application to suit user needs and potentially add/discard elements from the current version of our application. This iterative/agile development process would help us create a version that could be unanimously adopted in clinical practice.


Top of Form



Bottom of Form


Our usability test strategy, crafted for brevity at under 12 min, skillfully addressed our initial hypothesis with a well-balanced questionnaire of descriptive and quantitative questions. Results were stored and processed in a tabular spreadsheet to ensure data integrity. Utilizing cross-tabulation, we explored correlations among different response variables, revealing noteworthy associations, such as the positive correlation between the liking of the web application page in Figure 2 and the appeal for tabs in Figure 3 and Figure 4 or low TLX effort score consistent with high TLX performance. It is important to acknowledge the constraint of a small sample size, making it challenging to deem these findings as statistically significant. Additionally, we organized qualitative responses into thematic groups, uncovering recurring patterns, sentiments, and trends within open-ended questions. This approach facilitated the identification of outliers and unusual responses, offering valuable context and highlighting specific issues, as detailed in the raw feedback in Table 4. The primary outcome of our usability tests was the acquisition of key insights, the illumination of important trends, and the generation of actionable recommendations for future application iterations. To fortify the robustness of our analysis, we took the final step of validation by sharing our findings with human–computer interaction (HCI) experts.

## 4. Discussion

The comprehensive evaluation of our web application and survey results provides valuable insights into user experiences, shedding light on aspects such as ease of understanding, ease of use, and acceptability. NASA TLX Usability Test findings offer a quantitative assessment of workload metrics, contributing to a holistic understanding of our application’s usability. The feedback and quantitative data highlight that the majority of users found our web application’s user interface to be comprehensible and intuitive, with an average rating of 5.5 out of 7 on the Likert scale. However, a brief explanation was needed for some users, emphasizing the importance of clear instructions. Feedback regarding the donut chart representation indicates a need for improvement. Suggestions for optional activation of this representation and enhancing visual appeal will be considered.

Even though we endeavored to implement optimal practices in crafting an intuitive visualization, including the incorporation of presurvey instructions, a comprehensive legend, adherence to a conventional color scheme, and the integration of interactive hovers. To enhance the visualization’s efficacy, future iterations could benefit from supplementary clear explanations, the avoidance of technical jargon, the application of iterative design methodologies, and the solicitation of external feedback to introduce novel perspectives—a practice regrettably overlooked in the present evaluation. This holistic approach aims to fortify the robustness of our visualizations.

The unanimous preference for the detailed summary tab (rated 6.5 out of 7) reflects users’ desire for a streamlined workflow in busy clinical routines. Medical users would prioritize applications that provide precise information quickly, in addition to the value proposition of AI predictions. This feedback is valuable, and future iterations will eliminate irrelevant data, ensuring relevant features are easily accessible. The feedback related to patient demographic information and biopsy image resolution reasserts the importance of including relevant data and high-quality images for diagnostic value. The users’ consensus on the ease of navigation within the application reaffirms its user-friendliness. The highly positive reception of our web application among users, with all participants expressing a willingness to adopt it in their clinical workflows, underscores its potential value in healthcare settings and successfully validated our hypothesis of using multimodal data as follows:

The incorporation of patient EHR data offered valuable insights into the patient’s medical history, contributing to a more holistic approach to diagnosis and treatment planning despite all features not being equally paramount. High-resolution biopsy WSIs, when integrated into the grading process, provided detailed information about tissue structures, aiding in the identification of subtle patterns that could be critical for accurate grading. This also helped establish trust as at this stage pathologists had to crosscheck model predictions by looking at biopsies themselves. Finally, the application of report summaries served as a bridge between AI-generated predictions and clinician understanding. Concise summaries enhanced communication, ensuring that the insights provided are effectively translated into actionable information for clinical decision-making.

Our results also provided us valuable insight into the benefits and limitations of integrating AI-driven grading in the diagnosis of prostate cancer. Enhanced accuracy and efficiency possess high upside value. Human pathologists contribute extensive domain knowledge and interpretive skills, while AI algorithms excel in quickly analyzing large datasets, identifying subtle patterns or abnormalities. This collaborative approach allows for AI to automate routine tasks and prescreening, enabling pathologists to dedicate more time to complex cases, resulting in significant improvements with respect to diagnostic speed. Users appreciated the integration of AI and multimodal data coagulation, considering it an upgrade to traditional clinical workflows that involved referencing various files and reports for decision-making. Enhancing the quality of presented data in future iterations would solidify the value of this feature within our application. Other points for discussion are that AI systems can offer consistent evaluations, minimizing interobserver variability among human pathologists and ensuring standardized grading. Additionally, effective data retrieval can aid pathologists in managing information overload.

Limitations include the interpretability and trustworthiness of AI conclusions. Pathologists may be hesitant to fully rely on AI without a clear understanding of its grading decision rationale. Additionally, while AI is bound to excel in routine tasks, its limitations will become apparent in complex or rare cases where human pathologists can leverage their experience to navigate nuanced clinical contexts. Ethical and legal considerations, including issues of patient consent, data privacy, and potential biases in AI algorithms, will require thorough attention and iterative improvements to maintaining their relevance in the dynamic field of pathology.

The most positive aspect of our study was that users are open to utilizing AI-driven predictions to guide their diagnostic process. We aim to explore this feedback in-depth, presenting the relevance and accuracy of predictions effectively. The feedback regarding chart styles and color spectrum emphasizes the importance of visual clarity and user-friendliness, to be addressed in future iterations. The average rating of 5.5 out of 7 for acceptability and accessibility questions leans towards a winning cause.

NASA TLX Usability Test results indicate users generally perceived low to moderate levels of workload while interacting with the application. The highest median scores were associated with mental and temporal demand, categorizing them as having a medium level of difficulty. None of the categories reached high difficulty levels, suggesting users did not find any specific task particularly challenging. These findings are encouraging, indicating our application’s usability is generally in line with users’ expectations. We can be confident in our development if, in future iterations, average scores for all categories are much lower. Exploring other usability tests for more nuanced feedback is also under consideration.

## 5. Conclusions

This comprehensive study offers a profound understanding of user perspectives and experiences concerning our healthcare web application from five highly experienced medical practitioners. Our results illustrate that users appreciated the application’s clean and lucid presentation. Nonetheless, users proposed enhancements in the representation of charts and more nuanced information. The importance of presenting AI results in a comprehensible manner was underscored, especially for nontechnical users, implying the necessity of precise and concise explanations. Users generally found the application easy to navigate. Emphasizing the importance of consistency in data presentation and the need for better clarity in conveying AI results are major steps we will focus on based on feedback, as this is the core value proposition of our tool. The unanimous favoritism towards the detailed summary tab, supported by a high average rating of 6.5 out of 7, highlights its pivotal role in simplifying clinical workflows.

In the future, we aim to expand participant cohort to over 20 expert participants from locations around the world to gather more perspective and improved solutions. Collaborating with a panel of five subject matter experts provided invaluable insights into the evaluation of our innovative application. Nonetheless, the experimental design, constrained by a small sample size, posed significant challenges in obtaining robust inferential statistics. Attempts to estimate variance and confidence intervals were not useful, riddled with high variability and provided unreliable results due to the small sample size. In future endeavors, we will prioritize the adoption of more extensive statistical frameworks, facilitating the quantification of results with increased precision.

We strive to expand our data warehouse by finding more relevant datapoints for prostate cancer diagnosis, as suggested by participants. This could only be achieved by combined efforts, directly working with hospitals and medical centers to provide them the best insights from patient data available to them in a secure manner. The data we used were intentionally anonymous for patient privacy protection and could not possibly contain case numbers or detailed history. In the future, this framework can be integrated with Firebase for a much wider study that will collect stats on every click and help us profile participants based on their behavior. For example, how many clicks or how much time they spent on a single case, what tab they interacted with the most, how much time it takes for the image to upload and provide results, etc. This would open a completely new and exciting avenue for investigation, aiding in making more strategic design changes and improving the algorithm’s time and space complexity for the best performance in fluctuating internet connectivity. Our application, at this stage, did not have the capability to collect such digital footprints. NLP summaries were generated from data available acting as a hit-or-miss metric for specific diagnosis (prostate cancer grading in our case) in terms of relevance. Since the overall feedback for this section was deemed as “not useful”, more thought must be given for improving this feature or disregarding it all together in future versions. Exploring the integration of this application with existing healthcare systems and electronic health records (EHRs) could streamline clinical workflows further. Ensuring compatibility with various medical image formats and data sources is crucial for widespread adoption. For successful deployment of our application a major step in the future would be ensuring compliance with healthcare data privacy regulations (such as HIPAA in the United States) and obtaining necessary approvals from ethics boards for clinical use are essential steps. While specific applications and rules may vary, our foremost priority is to securely acquire and utilize patient data.

We are committed to thoroughly exploring and implementing software security standards that align with responsible clinical usage. Additionally, we remain prepared to address any unforeseen constraints that may emerge in subsequent iterations with the utmost emphasis on resolution.

To enhance scalability, we plan to integrate Amazon Web Services (AWS) with Docker for our web application. AWS’s Elastic Compute Cloud (EC2) instances will strategically host the Flask application, ensuring scalable computing capacity with autoscaling features. Docker and Docker Compose will facilitate multi-container orchestration. AWS Elastic Container Service (ECS) will streamline deployment with advanced features. AWS Fargate will optimize serverless container management. Integration with a PostgreSQL database container, orchestrated through ECS, will ensure a cohesive environment. This approach, coupled with AWS’s scalable infrastructure and Docker’s containerization, aims to achieve unparalleled scalability, dynamically adapting to evolving healthcare diagnostics demands. Anticipating a threefold increase in user capacity and a 25% reduction in deployment time, these enhancements envision a more robust and efficient system for future demands. Additionally, providing user training and support materials, such as tutorials or documentation, can help users fully utilize the application’s capabilities and make the most of its features.

As the field of AI in healthcare continues to evolve, ongoing research and discussion around topics like multimodal data integration, AI-aided diagnosis, and value assessment techniques will be instrumental in refining the integration of AI tools into clinical workflows, ultimately improving the diagnosis and management of prostate cancer. It is imperative to strike a balance between the strengths of AI and human expertise, ensuring that the combined approach optimally serves the needs of both healthcare providers and, most importantly, patients. Since the cost of false negatives can be catastrophic (predicting absence of cancer when it is actually present), solely relying on AI at this stage is out of the question; hence, building tools to aid practitioners is the way to go in the near future.

Overall, the feedback received was positive, constructive, and encouraging to keep developing better versions of this application. With this study, we could showcase that the application can definitely add value in clinical workflow and has the potential to be widely adopted.

## Figures and Tables

**Figure 1 cancers-15-05659-f001:**
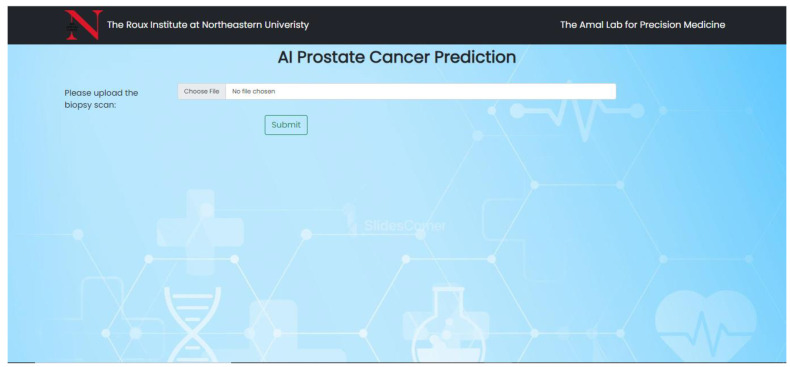
Web application: landing page.

**Figure 2 cancers-15-05659-f002:**
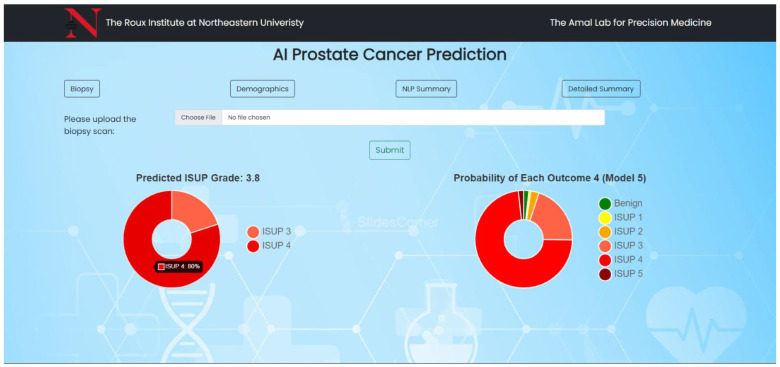
Web application: after a biopsy is successfully uploaded and processed.

**Figure 3 cancers-15-05659-f003:**
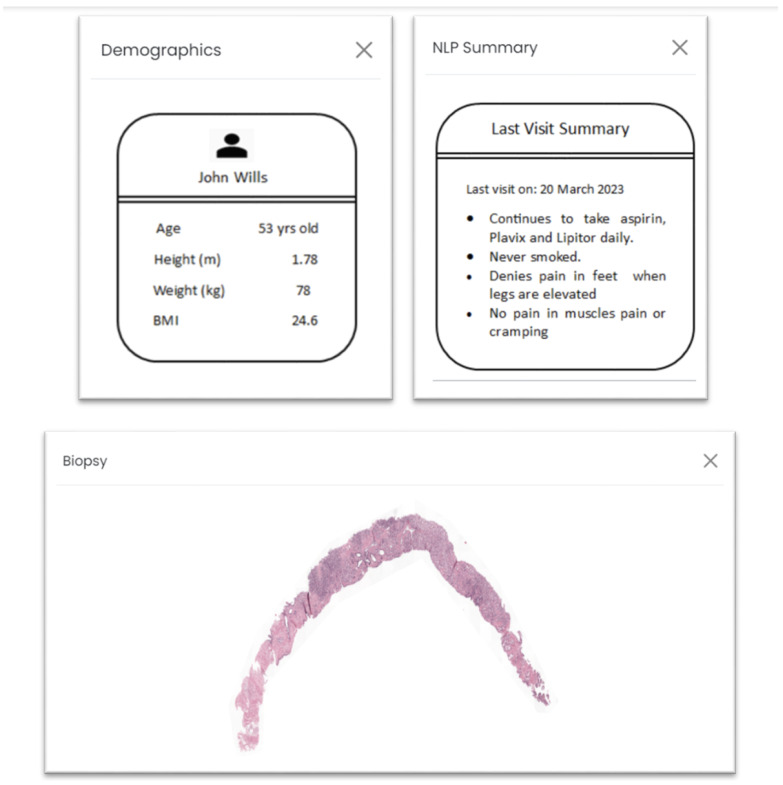
Demographics, NLP summary, and biopsy tab on the web application after a biopsy is successfully uploaded.

**Figure 4 cancers-15-05659-f004:**
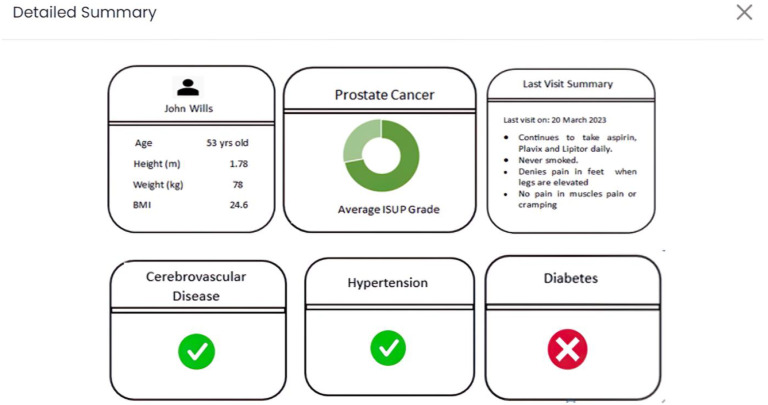
Detailed summary tab on the web application after a biopsy is successfully uploaded (application view).

**Table 1 cancers-15-05659-t001:** Deep learning model hyperparameters.

Model Hyperparameters	Value
Input image size	1024 × 1024
Dropout rate	0.3
Reduction ratio	16
Learning rate	1 × 10^−3^
Batch size	2
Epochs	20
Loss function	BCEWithLogitsLoss

**Table 2 cancers-15-05659-t002:** Section outline and number of questions based on question type.

Section Number	Section Name	Descriptive Questions	Rating-Based Questions	Multiple Choice Questions
1	Landing Page	1	1	0
2	Page After Biopsy Upload	1	10	0
3	Biopsy Tab	1	3	2
4	Demographics Tab	1	3	0
5	NLP Summary Tab	1	2	1
6	Overall Experience	0	7	0

**Table 3 cancers-15-05659-t003:** Average mean and median ratings of NASA TLX Usability questions by expert medical users.

NASA TLX Categories	Mean ($\bar{X}$) Score	Median ($\tilde{X}$) Score
Mental demand	7	9
Physical demand	3.5	4
Temporal demand	7.5	10
Effort level	4.5	4
Performance level	6.5	6
Frustration level	7	6

**Table 4 cancers-15-05659-t004:** Quoted feedback of participants on various usability domains and overall qualitative interpretation.

Themes	Sub-Themes	Feedback Quote	Overall Assessment
Ease of understanding	Web application presentation neatness, and simplicity	“Seems pretty clear. The left is giving the predicted grade based on the Gleason score. The right shows the ‘work’ behind the scenes to demonstrate how it go to that score”—User 4 “One the left side, have you ever considered using a vertical bar to represent different percentage of tissue, including benign one and cancers of different ISUP grades? My rationale is that the pathologists derive these mental percentage from a linear biopsy, your linear representation may make the mental transition more seamless and less frictional.”—User 3	Users generally found the web application’s presentation neat and clear. They appreciated the left side’s predicted grade based and the right side’s representation of the model’s decision-making process (refer Figure 2). Some users suggested using different charts to represent different percentages of tissue, including benign and cancerous tissue of various ISUP grades. This could potentially improve the mental transition when interpreting the results. There was a recommendation to include specific requirements for the images to be uploaded, this aimed to ensure data quality and consistency. In summary, the feedback highlighted the importance of clear visualization, potential enhancements to the presentation format, and the need for specific image upload requirements to improve the overall usability and effectiveness of the web application.
	AI result visualization charts and their self-explainable quality	“Should there be specifications for the image to be uploaded such as the magnification used for the slide scan, the model of the scanner, that it must be an H&E slide, that the slide must be deidentified and contain a date of the scan, etc.”—User 1	
Ease of usage	Clarity with instructions and navigation	“The data is easy to interpret on the right. I can see the breakdown of how the AI predicted the various scores and percent probability. There is a disconnect on the left because if it is using the data on the right for prediction, then it should simply state that the predicted score is 4 + 3 = 7 with Grade group of 3 and not 3.8”—User 4 “Easy to Navigate”—User 5	Users found the application relatively easy to use and navigate. To use this feedback to improve our application, we can ensure that the application maintains consistency in presenting data and that users can easily understand how predictions are made. We will also continue to gather user feedback and conduct usability testing to identify any other areas where improvements can be made. This will help in refining the user experience and making the application even more user-friendly. By implementing these actions, we can enhance the clarity and usability of our application, making it more effective and user-friendly for our target audience.
Accessibility	Positive feedback upon completion of web session	“It’s good. A little more detail on the Gleason score graphic would be useful. Caters to a wide healthcare audience.”—User 4 “Make the right column info optional: the user has to click a button for it to appear. I am not exactly certain on the details of the information in that column. If you explain to me further, I might have some ideas as how to make it user friendly.”—User 3	User feedback suggests that our application has been generally well received by healthcare professionals, with positive remarks about its quality and suitability for a broad audience. However, users have expressed a desire for more detailed information in the Gleason score graphic, indicating room for improvement in enhancing the clarity of predictions. To address this, we are considering making the right column information optional, requiring user interaction to access it unless we can transmit our intended results better to a non-technical audience. By noting down additional information aspects valuable for diagnosis, we will try our best to include suggested information by making changes to tabs as well as removing non-useful information based on received feedback. These insights from users will guide our ongoing efforts to refine and optimize our application, ultimately delivering a more robust and user-friendly tool for healthcare practitioners.
	Difficulties faced during interaction	“Information on family history of prostate cancer not present, useful for diagnosis”—User 2 “Not very comprehensive (don’t know about prior biopsies or PSA level)”—User 5	
Tab-specific value proposition	Biopsy Tab	“Would be better to see the biopsy in a separate image viewer or management system.”—User 4 “Looking at the biopsy is very helpful”—User 5	Key takeaways from user feedback indicate a preference for a dedicated image viewer or management system for biopsy viewing with higher image resolutions. Users find the biopsy examination to be particularly valuable. On the Demographics Tab, there is a consensus that more comprehensive demographic data, apart from age, should be included to assess potential risk effectively. However, users emphasize excluding unrelated risk factors. Regarding the NLP Summary Tab, it is suggested that its current content does not significantly contribute to prostate diagnosis and grading, indicating room for improvement in its relevance. Lastly, users express a strong preference for viewing all information together in the Detailed Summary Tab, emphasizing the need for a streamlined and efficient presentation. To enhance our application, we plan to explore dedicated biopsy viewing options, incorporate more pertinent demographic data while excluding irrelevant factors, refine/disregard the NLP Summary Tab for better relevance, and optimize the presentation of data in the Detailed Summary Tab.
	Demographics Tab	“Insufficient demographics shown to assess potential risk”—User 1 “Other than age, other risk factors should not be here.”—User 3	
	NLP Summary Tab	“Not very helpful for prostate diagnosis/grading”—User 5 “OK to know but does not really lend benefit for the current study”—User 3	
	Detailed Summary Tab	“Better to see all information together (Faster)”—User 5 “Yes, I’d prefer to see all this information together Yes, it doesn’t seem overwhelming.”—User 2	
Personalized suggestions regarding interface	Optional suggestions provided by each user	“Helpful tool”—User 1 “No strong suggestions for improvement at the moment”—User 2 “It provides too much information. There are only 3 things crucial to the pathologists: whether there is cancer, if yes where are they on the biopsy, and what are the grades.”—User 3 “Minor point but the light blue background does not seem to fit. It takes attention away from the visuals and should either be completely light or darkened without the background images.”—User 4 “Nice design/visually appealing”—User 5	Feedback on the personalized suggestions for the interface is generally positive from each user, with users finding the tool helpful and visually appealing. However, there is a suggestion to streamline the information presented, focusing on key elements crucial to pathologists. Additionally, a minor design adjustment is recommended to improve the background’s visual coherence. To enhance our application based on this feedback, we will work on optimizing the information presented for relevance and clarity, while also addressing the design element to create a more visually cohesive experience for users.

## Data Availability

The public dataset used for modeling our deep learning algorithm can be found on the Kaggle website. Survey results presented in this study are available on request from the corresponding author.

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
