# Peer review of "Enhancing Prostate Cancer Diagnosis with a Novel Artificial Intelligence-Based Web Application: Synergizing Deep Learning Models, Multimodal Data, and Insights from Usability Study with Pathologists"

_cancers, 2023, doi:10.3390/cancers15235659_

Round 1

Reviewer 1 Report

Comments and Suggestions for Authors

This study provides a very interesting auxiliary tool. This study mainly focuses on ease of understanding, ease of use, and acceptability. It is hoped that the team can provide more functions and data on diagnostic accuracy and treatment prediction in the future system.

Author Response

Please see attachment with addressed comments followed by manuscript with highlighted changes

Reviewer 2 Report

Comments and Suggestions for Authors

I have read the paper on this page and here are three areas that could be improved: Introduction: The introduction is too long and contains some redundant or irrelevant information. For example, the paragraph on multimodal data sources does not clearly relate to the main topic of the paper, which is prostate cancer diagnosis using a web application. The introduction is improbable to be shortened and focused on providing the background, motivation, and objectives of the study. Methodology: The methodology section lacks some details and clarity on how the web application was developed and evaluated. For example, it does not explain how the EHR data was curated and interconnected with the biopsy images, or how the NLP summaries were generated1. It also does not describe the survey design and questions, or how the TLX NASA Usability Test was administered and scored. The methodology section could probably be expanded and organized to provide more information on the data sources, the web application features, and the evaluation methods. Outcomes: The outcomes section chiefly presents descriptive statistics and qualitative feedback from the survey, but does not provide any inferential statistics or comparisons with other tools or methods. For example, it does not offer any confidence intervals, p-values, or effect sizes to assess the significance and magnitude of the findings. It also does not compare the performance or usability of the web application with existing prostate cancer grading systems or standards. The results section could be enhanced by adding more quantitative analysis and benchmarks to support the claims and conclusions of the study.

Comments on the Quality of English Language

Based on the provided article entitled "Enhancing the Optimization of Prostate Cancer Diagnosis: Insights from Pathologists and Usability Survey Data on the Integration of Human Expertise and Artificial Intelligence Outcomes in a Web Application," the level of English proficiency exhibited is remarkably impressive. Throughout the article, the authors demonstrate an authoritarian comprehension of the English language, using specialized terminology, intricate sentence structures, and sophisticated expressions to effectively communicate their ideas. The written discourse is both lucid and succinct, with the authors demonstrating a commendable comprehension of academic writing conventions, including the judicious use of appropriate citations and references. Furthermore, the authors employ a variety of techniques to improve the readability and clarity, such as the inclusion of headings, bullet points, and summaries to break up the text and highlight important points. The article's organizational structure is cogent, featuring a distinct introduction, methodology section, results section, and conclusion. Despite this, there are a couple of trivial faults in the article that could be fixed. For instance, there are occasional instances of inelegant phrasing or incomplete sentences, and certain citations and references could benefit from enhanced consistency in formatting. Furthermore, several paragraphs might potentially be condensed and fine-tuned for heightened precision and comprehensibility, thus making the text more accessible and easily understood. In aggregate, the exceptional English skill exhibited in this article is highly praiseworthy, and the authors have skillfully presented their research in a lucid and impactful manner. With a few minor adjustments, the article could be even more robust and sleek.

Author Response

See attachment below that contains response to each comment followed by manuscript with highlighted changes

Reviewer 3 Report

Comments and Suggestions for Authors

In this manuscript the authors develop a web application to support prostate cancer grading by pathologists, integrating AI predictions, biopsy images, and patient data. Five pathologists tested the application and provided feedback via a survey and NASA TLX usability questionnaires. Results indicate the application was easy to understand and use. The visualizations were appreciated but some modifications could improve intuitiveness. All users were willing to adopt the tool in clinical workflows, citing the potential to reduce workload. TLX ratings showed a low-moderate mental workload, suggesting room for improving explanations and visuals.

No quantitative usability metrics were tracked during use, a limitation of the current study. The small sample size of 5 reviewers limits the generalizability of the usability feedback.

I have a few minor comments:

1.       Were any quantitative metrics tracked during use, like time to complete tasks or errors made, to complement the survey data?

2.       Does the small sample size of 5 reviewers (A panel of four experienced pathologists and one medical expert) limit the generalizability of the usability feedback? Increase reviewer sample size and diversity in future studies.

3.       How will the authors prioritize which feedback to implement in the next design iteration?

4.       What are the next steps for clinically validating the tool before deployment?

Author Response

(The authors gave the same response as above.)

Reviewer 4 Report

Comments and Suggestions for Authors

I am really sorry to state that this manuscript has nothing in common with a scientific manuscript and it seems to be more a manual or a guideline than even a report. The figures are included in the methodology and those shows just some screenshots of the application, while the results are included just the results of the survey. I highly recommend rejecting it, because it is not serious at all.

Author Response

(The authors gave the same response as above.)

Round 2

Reviewer 2 Report

Comments and Suggestions for Authors

An area that demands attention pertains to the section dedicated to the dataset and implementation details. Regrettably, the brevity of this section hinders the comprehensive understanding of the developmental processes underlying the web application, as well as the training and evaluation methodologies applied to the deep learning models. Noteworthy points that warrant further elucidation include:

- Elaboration on the acquisition and processing methodology employed for the Electronic Health Record (EHR) data sample, along with insights into the data integration process with the PANDA imaging dataset.

- Detailed exposition on the specific hyperparameters and training configurations adopted for the EfficientNet-B1 models, alongside a comprehensive account of the ensemble techniques employed and the corresponding validation procedures undertaken.

- In-depth exploration of the implementation of the Flask framework in the creation of the web application, accompanied by a thorough discussion of the technical challenges encountered and the subsequent resolutions employed in integrating the diverse data sources and visualizations.

- Comprehensive insights into the measures undertaken to ensure the scalability and performance of the web application, inclusive of a detailed account of the computational resources and platforms utilized during the developmental and deployment phases.

By enhancing this section with comprehensive elucidations and detailed explanations, the paper can significantly enhance its overall clarity, reproducibility, and scholarly credibility.

Comments on the Quality of English Language

The paper exhibits a well-structured framework adhering to the established format of a scientific article, encompassing an abstract, introduction, materials and methods, results, and discussion sections. The adept utilization of appropriate academic language and terminology, coupled with an absence of grammatical errors and spelling inaccuracies, contributes to the paper's commendable linguistic proficiency. The paper extensively references pertinent sources while maintaining a consistent and rigorous citation style throughout.

However, the paper encounters certain minor challenges concerning clarity and coherence, notably characterized by the deployment of ambiguous expressions, redundancies, and the absence of smooth transitions between paragraphs. For instance, in the abstract, the phrase "This makes sure the tool is really helpful for real medical needs" could benefit from a more specific elucidation of the tool's tangible benefits or impact, enhancing the overall precision of the statement¹. Furthermore, in the materials and methods section, the paragraph commencing with "The imaging dataset employed in this research initiative originates from the Prostate Cancer Grade Assessment Challenge (PANDA)" lacks a definitive transition from the preceding paragraph, necessitating the inclusion of a clear topic sentence to introduce the paragraph's primary theme.

To augment readability and succinctness, the paper could employ shorter sentence structures, bullet points, or tables to present data or information in a more organized manner. For instance, in the introduction, the lengthy sentence "Our web platform bridges this divide, combining human expertise with AI-driven grading using diverse data sources" could be effectively bifurcated into two concise sentences to enhance comprehension³. Moreover, in the materials and methods section, the sentence "The reaction, they claim, transformed the mixture into a dark gray, superconductive material" could be streamlined by adopting a direct voice and eliminating the phrase "they claim," thereby fostering a more authoritative tone within the discourse.

In the results section, the sentence "The survey was divided into six primary sections, aiming to collect detailed input on each aspect of the web application" could be rendered more accessible through the incorporation of a succinct bullet point list delineating the six sections and their respective objectives, thereby fostering enhanced readability and comprehension⁴. Overall, while the paper exhibits a commendable level of English proficiency, it would benefit from certain revisions to fortify its clarity, coherence, and conciseness.

Author Response

Thank you for second review round. We dearly appreciate the effort you have made to help improve our manuscript.

Please see attachment below.

Reviewer 4 Report

Comments and Suggestions for Authors

The manuscript from the co-authors have been improved significatively, though it was difficult to me to understand this kind of manuscript, which is totally different from a traditional scientific manuscript. I would recommend the Editor in Chief of the Journal to accept the manuscript in the presented form.

Author Response

Thank you for taking out the time to review our manuscript. We really appreciate your feedback and are glad to know that you are content with the latest (updated) version. 

Round 3

Reviewer 2 Report

Comments and Suggestions for Authors
  1. To strengthen the logical persuasion of this paper's argument, three points that could be supplemented are:

  2. a) Further discussion on the specific benefits and limitations of integrating human expertise and AI-driven grading in the diagnosis of prostate cancer.
    b) In-depth analysis of the usability test results, including specific areas of improvement for explanations and data visualization.
    c) Exploration of how the application of multimodal data sources, such as patient EHR data, high-resolution biopsy WSIs, and report summaries, enhances the accuracy and efficiency of prostate cancer grading.
  3.  
  4. The paper successfully addresses the importance of optimizing prostate cancer diagnosis by integrating human expertise and AI outcomes through a web application. It highlights the positive feedback received from pathologists and medical practitioners, indicating the usability and potential value of the developed tool in clinical settings. Additionally, the inclusion of diverse data sources and the use of deep learning techniques demonstrate the thoroughness of the research.

Author Response

Please see attachment below, Thank you for working with us to improve the paper, we appreciate all the feedback
